# Tas-102 for Refractory Metastatic Colorectal Cancer: A Multicenter Retrospective Cohort Study

**DOI:** 10.3390/cancers15133465

**Published:** 2023-07-02

**Authors:** Matteo Conti, Elena Bolzacchini, Giovanna Luchena, Lorenza Bertu’, Paola Tagliabue, Stefania Aglione, Antonio Ardizzoia, Jessica Arnoffi, Francesco Maria Guida, Alessandro Bertolini, Alessandro Pastorini, Maria Duro, Donato Bettega, Giovambattista Roda’, Salvatore Artale, Alessandro Squizzato, Monica Giordano

**Affiliations:** 1Oncology Unit, Sant’Anna Hospital, ASST Lariana, 22042 San Fermo della Battaglia, Italy; matteo.conti2007@libero.it (M.C.); giovanna.luchena@asst-lariana.it (G.L.); monica.giordano@asst-lariana.it (M.G.); 2Department of Medicine and Surgery, University of Insubria, 21100 Varese, Italy; lorenza.bertu@uninsubria.it (L.B.); alessandro.squizzato@asst-lariana.it (A.S.); 3Oncology Unit, Vimercate Hospital, ASST della Brianza, 20871 Vimercate, Italy; paola.tagliabue@asst-vimercate.it (P.T.); stefania.aglione@asst-vimercate.it (S.A.); salvatore.artale@asst-vimercate.it (S.A.); 4Oncology Unit, Alessandro Manzoni Hospital, ASST Lecco, 23900 Lecco, Italy; a.ardizzoia@asst-lecco.it (A.A.); j.arnoffi@asst-lecco.it (J.A.); f.guida@asst-lecco.it (F.M.G.); 5Oncology Unit, Ospedale Civile di Sondrio, ASST Valtellinese, 23100 Sondrio, Italy; alessandro.bertolini@asst-val.it (A.B.); alessandro.pastorini@asst-val.it (A.P.); 6Oncology Unit, Valduce Hospital, 22100 Como, Italy; mduro@valduce.it; 7Oncology Unit, Sacra Famiglia Hospital, 22036 Erba, Italy; dbettega@fatebenefratelli.eu; 8Oncology Unit, Istituto Clinico Humanitas Mater Domini, 21100 Castellanza, Italy; gianni.roda@gmail.com; 9Department of Internal Medicine, Ospedale Sant’Anna, ASST Lariana, 22100 Como, Italy

**Keywords:** TAS-102, mCRC, DPYD, HIV, elderly

## Abstract

**Simple Summary:**

Trifluridine/tipiracil (TAS-102) is an oral chemotherapy approved for the treatment of metastatic colorectal cancer (mCRC). Efficacy and safety of TAS-102 was shown in phase II-III clinical trials and in several real-life studies but elderly and other special subgroups are underrepresented in clinical trials. The aim of our study is to evaluate the effectiveness and safety of TAS-102 in consecutive patients with pretreated mCRC treated in a real-life Italian large cohort. Our study confirms the effectiveness and safety of TAS-102 in patients with pretreated mCRC, suggesting a similar risk-benefit profile in the elderly.

**Abstract:**

Trifluridine/tipiracil (TAS-102) is an oral chemotherapy approved for the treatment of metastatic colorectal cancer. The efficacy and tolerability of TAS-102 were shown in phase II-III clinical trials and in several real-life studies. The elderly and other special subgroups are underrepresented in published literature. We conducted a retrospective multicenter study to assess the effectiveness and safety of TAS-102 in consecutive patients with pretreated mCRC. In particular, we estimated the effectiveness and safety of TAS-102 in elderly patients (aged ≥70, ≥75 and ≥80 years) and in special subgroups, e.g., patients with concomitant heart disease. One hundred and sixty patients were enrolled. In particular, 71 patients (44%) were 70 years of age or older, 50 (31%) were 75 years of age or older, and 23 (14%) were 80 years of age or older. 19 patients (12%) had a concomitant chronic heart disease, three (2%) patients were HIV positive, and one (<1%) patient had a *DPYD* gene polymorphism. In 115 (72%) cases TAS-102 was administered as a third-line treatment. The median overall survival (OS) in the overall population was 8 months (95% confidence interval [CI], 6–9), while the median progression-free survival (PFS) was 3 months (95% CI, 3–4). No significant age-related reduction in effectiveness was observed in the subpopulations of elderly patients included. The toxicity profile was acceptable in both the whole and subgroups’ population. Our study confirms the effectiveness and safety of TAS-102 in patients with pretreated mCRC, suggesting a similar risk-benefit profile in the elderly.

## 1. Introduction

Colorectal cancer (CRC) is the most common form of malignancy in the gastrointestinal (GI) tract [1]. New treatment options, such as TAS-102, have prolonged survival also for patients with metastatic CRC (mCRC) [2]. TAS-102 is the combination of trifluridine (tri-fluorothymidine), a nucleoside analog of thymidine, and tipiracil hydrochloride. After the uptake in neoplastic cells, trifluridine is incorporated into DNA in its active triphosphate form and interferes with neoplastic cells proliferation, while tipiracil hydrochloride maintains high plasma concentrations of trifluridine by inhibiting the enzyme responsible for its degradation (trifluridine phosphorylase) [3]. TAS-102 was approved for the treatment of metastatic CRC beyond second-line treatment following the phase III RECOURSE trial, which showed a higher median overall survival (OS) and median progression-free survival (PFS) in the TAS-102 group compared to the placebo group [4]. The different mechanisms of action and degradation between TAS-102 and the other fluoropyrimidine analogs in CRC therapy seem to explain the different cardiotoxicity profiles (i.e., cardiac arrythmias, miocardial infarction and angina-like symptoms) of trifluridine/tipiracil, which was found to be between 2 and 40 times less cardiotoxic than 5-fluorouracil (FU) in an analysis of phase I, II and III clinical trials [5]. Indeed, TAS-102 is not catabolized by dihydropyridine dehydrogenase (DPD), the main enzyme involved in the degradation of 5-FU and capecitabine, but by thymidine phosphorylase (TPase), thus resulting in a reduced formation of fluoroacetate, the main cardiotoxic metabolite in the fluoropyrimidine degradation pathway [5]. This evidence is also confirmed by the low incidence of cardiovascular adverse effects reported in several observational studies [6,7,8,9].

Although an increasing number of elderly patients with also comorbidities require cancer treatments in advanced lines, these patients are often underrepresented in clinical trials. Indeed, a subgroup analysis of the RECOURSE study confirmed the efficacy and safety of TAS-102 even in elderly patients, both considering age 65 and age 70 years as cut-off [10]. Furthermore, an Italian study conducted on 50 elderly patients (aged > 70 years) showed a similar risk-benefit profile with improved quality of life [11].

Polymorphisms resulting in dihydropyrimidine dehydrogenase (DPD) activity reduction can cause grade 3 or 4 gastrointestinal, hematological, and hand-foot toxicities after administration of 5-FU and capecitabine, being DPD the main enzyme implied in the catabolism of these drugs, as mentioned before. As TAS-102 degradation follows another degradation pathway, TAS-102 may also be safe in patients with *DPYD* polymorphisms [12,13]. Finally, since leukopenia and neutropenia are the most frequent toxicities of trifluridine/tipiracil demonstrated in both clinical trials and real-life studies [6,8,9,14,15], it would be clinically relevant to evaluate the safety of TAS-102 in patients with immunodeficiency-related comorbidities, e.g., HIV positive, for which there are no available efficacy or safety data in the literature.

The aim of this retrospective multicenter cohort study was to assess the effectiveness and safety of TAS-102 in a real-life context. In particular, we planned to assess the effectiveness and safety of TAS-102 in elderly patients and in some special subgroups, such as patients with concomitant heart diseases, *DPYD* polymorphisms or HIV positive, who are underrepresented in published literature.

## 2. Materials and Methods

We retrospectively enrolled all patients with metastatic CRC (mCRC) consecutively treated with trifluridine/tipiracil (TAS-102) from August 2015 to May 2022 in seven oncology Units of Lombardy, Italy. The inclusion criteria were the following: objective (i.e., histological and radiological) diagnosis of mCRC, ineligibility to standard chemotherapy (i.e fluoropyrimidines, irinotecan, oxaliplatin, anti-vascular endothelial growth factor monoclonal antibody, and epidermal growth factor receptor monoclonal antibodies for RAS wild-type tumors) or chemo-refractory disease (i.e., disease progressed under standard treatments in previous lines of therapy), and administration of at least one dose of trifluridine/tipiracil (TAS-102).

Data regarding age, sex, TNM staging at the time of diagnosis, Eastern Cooperative Oncology Group Performance Status (ECOG PS), tumor sidedness, number and localization of metastatic sites, *RAS (KRAS* and *NRAS)* and *BRAF* mutation profile, microsatellite stability (MSS) or instability (MSI), previous oncological therapies, the time between diagnosis of metastatic disease and first administration of TAS-102, *DPYD* status, history of heart disease and HIV infection were collected.

Mutational status analysis of *KRAS*, *NRAS* and *BRAF* genes, and evaluation of the stability status of microsatellites and *DPYD* polymorphisms were locally performed by experienced specialists at certified molecular genetics laboratories. The allelic variants of the *DPYD* gene sought by PCR analysis were: *DPYD*2A* (IVS14-1G > A, c.1905 + 1G > A, rs3918290), *DPYD*13* (p.I560S, c.1679T > G, rs55886062), DPYD D949V (p.D949V, c.2846A > T, rs67376798) and *DPYD IVS10* (c.1129-5923C > G, rs75017182).

A positive history of heart disease was defined as previous ischemic heart disease, heart failure, and arrhythmias.

Patients were periodically screened for toxicities through clinical evaluation and laboratory assessments according to local clinical practice. Grading of adverse events (AEs) was assessed according to Common Terminology Criteria for Adverse Events (CTCAE) v.5.0. [16]. Disease response was evaluated through physical examination and radiologic assessment (computed tomography scan, positive emission tomography scan or magnetic resonance imaging) performed periodically according to the best clinical practice. Radiologic assessments were evaluated following RECIST 1.1 criteria [17].

TAS-102 was administered orally twice a day in 28-day cycles (five days of treatment and subsequently two days of rest in the first two weeks of the cycle, followed by a 2-week rest period) at the starting dose of 35 mg/m^2^.

The primary endpoint of the study was to evaluate the effectiveness of TAS-102 in terms of OS and PFS. The secondary endpoints were: (i) to assess OS and PFS in elderly patients (cut-offs: 70, 75 and 80 years); (ii) to assess OS and PFS in special subgroups (i.e., patients affected by heart disease, HIV positive, or *DPYD* polymorphisms); iii) to assess the safety of TAS-102 in terms of adverse effects (AEs) in the whole study population, in the elderly, and in the subgroups.

The study protocol was approved by Asst Settelaghi ethics committee (n 114/2023).

### Statistical Analysis

Continuous variables are expressed as median and range; categorical variables are presented as absolute and relative frequencies.

OS and PFS curves were constructed and the median OS and PFS along with their 95% confidence intervals (Cis) were calculated. The time between the start of treatment with TAS-102 and the death of the patient (or the end of follow-up for patients alive at the time of analysis) was considered to calculate OS. For the calculation of PFS, the time interval between the initiation of TAS-102 therapy and the first of the following events was considered: radiological or clinical progression, death, exit from follow-up.

A univariate Cox proportional hazard model was applied in the overall study population, to explore the association with OS of the following variables: sex, age, site of primary tumor, Performance Status ECOG, mutational status of *RAS* and *BRAF*, microsatellite stability (MSI/MSS), staging of disease at onset, number of previous lines of treatment, previous treatment with regorafenib, mono- or multi-metastatic disease, the time between diagnosis of metastasis and initiation of TAS-102 therapy, presence of liver, lung and peritoneal metastasis. The hazard ratios (HR) with a 95% confidence interval and their *p*-value have been calculated. The comparison between the Kaplan–Meier curves for OS and PFS for the different age cut-offs (70, 75 and 80 years) was performed by log-rank test.

To study the association between age, at different cut-offs, and the presence of toxicity, a chi-square test or Fisher’s exact test was used in case of expected values below the threshold frequency of 5.

*p*-values below 0.05 were considered to indicate statistical significance.

The follow-up deadline for data analysis was May 2022. The analyses were carried out using the SAS v.9.4 software, the graphs were built using the R (survminer package) software.

## 3. Results

### 3.1. Study Population

One hundred and sixty patients, 103 (64%) males and 57 (36%) females with mCRC treated between August 2015 and May 2022 receiving at least one dose of TAS-102 were enrolled in this study. Baseline characteristics are reported in Table 1. In particular, the median age of the study population was 68 years (range: 38–89), of which 71 patients (44%) were 70 years old or older, 50 (31%) were 75 years old or older, and 23 (14%) were 80 years old or older. The most frequent localization was the left colon (*n* = 71, 44%). Almost all patients had received fluoropyrimidine as part of a prior chemotherapy regimen (*n* = 158, 99%), in 146 (91%) cases in combination with oxaliplatin and in 147 (92%) cases in combination with irinotecan, 122 (76%) patients had received bevacizumab, 30 (19%) aflibercept, 53 (33%) anti-EGFR treatment (cetuximab and/or panitumumab) and 10 (6%) regorafenib prior to TAS-102 treatment. In 115 cases (72%) TAS-102 was administered as a third-line treatment. The mutation frequencies of *RAS* and *BRAF* were 56% and 3%, respectively, and the instability status of microsatellites (MSI) affected four patients (3%); 85 (53%) of patients received the diagnosis of metastatic disease, and the time between diagnosis of metastases and initiation of TAS-102 treatment was greater than 18 months for 113 patients (71%). Forty-four (27%) patients had one metastatic site, the remaining 116 (73%) had plurimetastatic disease. One hundred (63%) patients had an ECOG Performance Status of 1 at the time of TAS-102 treatment, 47 (29%) had an ECOG PS of 0 and 13 (8%) had 2 or more, including two patients (1%) with ECOG PS 3. Nineteen patients (12%) had a positive history of heart disease. One patient (<1%) had a *DPYD* gene polymorphism and three (2%) patients were HIV positive. *DPYD* gene polymorphism and HIV tests were performed in 21 patients and 111 patients, respectively.

### 3.2. Effectiveness

A total of 156 and 160 patients were included in the OS and PFS analysis, respectively. The median OS was 8 months (95% CI, 6–9) for a median number of TAS-102 cycles per patient of 3 (range 1–17). (See Figure 1a)

The median PFS was 3 months (95% CI, 3–4) (See Figure 1b)

In the univariate analysis, the group with ECOG PS 1 had a Hazard Ratio (HR) of 1.73 (95% CI, 1.17–2.55; *p*-value 0.01) for OS compared to ECOG PS 0. The group with ECOG PS ≥2 had HR of 2.21 (95% CI, 1.10–4.09; *p*-value 0.02) for OS compared to ECOG PS 0. (See Table 2)

### 3.3. Safety

One-hundred and forty (88%) patients reported AEs. Hematologic AEs were the most frequent: 78 (49%) patients developed neutropenia, 5 (3%) febrile neutropenia, 46 (29%) leukopenia (29%), 36 (23%) anemia, and 5 (3%) thrombocytopenia. Among non-hematologic AEs, the most frequent were asthenia (*n* = 67, 42%), diarrhea (*n* = 23, 14%), nausea (*n* = 20, 12%) and vomiting (*n* = 10, 6%). Hepatotoxicity was developed in six (4%) of patients. Finally, three (2%) patients had mucositis, two (1%) skin rash, declivous edema, gastrointestinal bleeding, or constipation, while nephrotoxicity (increased creatinine above baseline values), itching, alopecia, inappetence, dysgeusia, pneumonia, hypoglycemia, paresthesias, and headache were found in less than 1% of cases. Only one (<1%) patient reported cardiovascular toxicity, i.e., hypotension, during treatment with TAS-102. Data about toxicities grading were not available for the whole study population (not available for 83 patients, 52%). Among the data collected, considering the most severe adverse effects (G3–G4), neutropenia was the most frequently represented (*n* = 30, 19%); others included anemia (*n* = 7, 4%), febrile neutropenia and leukopenia (*n* = 5, 3%), asthenia (*n* = 3, 2%) and hepatotoxicity (*n* = 1, <1%). (See Table 3)

Adverse effects were the cause of treatment discontinuation for three (2%) patients. The first reported G3-grade neutropenia, G2-grade leukopenia, G1-grade anemia, pruritus, and diarrhea, which were considered not tolerable by the patient. The second patient discontinued treatment due to a drug-induced liver injury with the elevation of gamma glutamyl transpeptidase (grade G4), alkaline phosphatase (grade G3) and aspartate aminotransferase (grade G1). Finally, the third patient discontinued treatment due to the appearance of G4-grade febrile neutropenia, which required hospitalization.

After discontinuation of TAS-102 treatment, 41 (25.6%) patients were considered fit for the administration of one or more subsequent lines of treatment, i.e., regorafenib, capecitabine, mitomycin, panitumumab, cetuximab, rechallenge with previous lines therapy, chemoembolization or radiotherapy of metastatic sites.

### 3.4. Elderly Patients

The median OS was 8 months (95% CI, 6–10) in patients ≥ 70 years old and 7 months (95% CI, 6–9) in patients < 70 years old (*p* = 0.9). (see Figure 2a)

The median PFS was 4 months (95% CI, 3–5) in patients ≥ 70 years old and 3 months (95% CI, 3–4) in patients < 70 years old (*p* = 0.42). (see Figure 2b)

The median OS was 8 months (95% CI, 6–11) in patients ≥ 75 years old and 7 months in patients < 75 years old (*p*= 0.6). (see Figure 3a)

The median PFS was 4 months (95% CI, 3–5) in patients ≥ 75 years old and 3 months (95% CI, 3–4) in patients < 75 years old (*p* = 0.14). (see Figure 3b).

The median OS was 8 months (95% CI, lower limit 6) in patients ≥ 80 years old and 7 months (95% CI, 6–9) in patients < 80 years old (*p* = 0.25). (see Figure 4a)

The median PFS was 5 months (95% CI, lower limit 3) in patients ≥ 80 years old and 3 months (95% CI, 3–4) in patients < 80 years old (*p* = 0.042). (see Figure 4b)

The toxicity profile was similar in the three subgroups of elderly patients analyzed, with hematologic adverse effects predominating, mainly in the form of neutropenia (*n* = 37, 52.1% of the population ≥ 70 years old) (see Table 4). The most frequent non-hematologic toxicity was asthenia (*n* = 27, 38% of the population ≥ 70 years old) followed by gastrointestinal toxicities (diarrhea *n* = 9, 12.7%; nausea *n* = 5, 7%; vomiting *n* = 2, 2.8%; percentages referred to the population ≥ 70 years old). Neutropenia was the most frequent toxicity even when considering grade G3 or G4 adverse events (*n* = 14, 25.4% of the population ≥ 70 years old excluding the subjects with unknown grading of AEs). No significant difference in terms of adverse effects was found comparing patients aged 70 years or older with the population younger than 70 years, whereas the comparison between patients aged 75 years or older with the population younger than 75 years revealed statistically significant differences in neutropenia, (*n* = 31, 62% of the population ≥ 75 years old vs. *n* = 47, 42.7% of the population < 75 years old; *p* = 0.02), febrile neutropenia (*n* = 4, 8% of the population ≥ 75 years old vs. *n* = 1, 0.9% of the population < 75 years old; *p* = 0.04), and asthenia (*n* = 15, 30% of the population ≥ 75 years old vs. *n* = 52, 47.3% of the population < 75 years old; *p* = 0.04). Comparing the population aged 80 years or older there were significant differences with the population aged less than 80 years in terms of grade G3 or G4 neutropenia (*n* = 8, 44.4% of the population ≥ 80 years old vs. *n* = 22, 20.4% of the population < 80 years old, *p* = 0.03; percentages calculated excluding the subjects with unknown grading of AEs), and of asthenia (*n* = 5, 21.7% of the population ≥ 80 years old vs. *n* = 62, 45.3% of the population < 80 years old; *p* = 0.03).

Only one (<1%) patient, aged 76 years, discontinued treatment following the occurrence of febrile neutropenia as AE.

### 3.5. Special Subgroups

#### 3.5.1. Heart Disease

Nineteen patients (12%) had previous heart disease. These patients did not develop any cardiovascular AEs during the treatment. Reported AEs were hematologic toxicity mainly in the form of neutropenia (*n* = 9, 47% of the subgroup), asthenia (*n* = 11, 58% of the subgroup), diarrhea (*n* = 2, 11% of the subgroup), and nausea (*n* = 2, 11% of the subgroup). None of these patients required discontinuation of treatment due to adverse effects.

#### 3.5.2. HIV+

Three (2%) patients were HIV positive, 108 (68%) were HIV−, while the remaining 49 (30%) were not tested for HIV status.

The first HIV+ patient completed eight cycles of TAS-102 and developed neutropenia and diarrhea as adverse effects during the third cycle; the second one completed three cycles of TAS-102 and developed febrile neutropenia; the third one completed three cycles of TAS-102 and had no adverse effects. None of these patients discontinued treatment due to toxicity.

#### 3.5.3. *DPYD* Gene Polymorphism

Only one patient with *DPYD* gene polymorphism, i.e., IVS10 mutation in heterozygosity, underwent treatment with TAS-102, 21 (13%) patients were wild type, while the remaining 138 patients (86%) were not tested for the *DPYD* gene polymorphism.

The patient with *DPYD gene* polymorphism received a total of five cycles of therapy, at a reduced dose due to baseline glomerular filtration rate less than 30 mL/min. During the treatment, G2 leukopenia, G1 neutropenia, and G3 asthenia developed. The patient did not require further dosage reductions or discontinuation of therapy due to toxicities.

## 4. Discussion

Our study confirms the effectiveness and safety of TAS-102 in patients with pretreated mCRC, suggesting a similar risk-benefit profile in the elderly and in some special subgroups.

Indeed, the primary aim of our study was to evaluate the impact of trifluridine/tipiracil on overall survival and progression-free survival in the context of routine clinical practice. In our cohort, the median OS was 8 months (95% CI, 6–9) compared with 7.1 months in the RECOURSE trial (95% CI, 6.5 months–7.8 months) [4]. Similar results were obtained in other European real-life cohorts: Carriles et al. [7] and Stavraka et al. [9]) showed a median OS of 8.3 months (95% CI, 6.2 months–9.87 months) and 7.6 months (95% CI, 6.5 months–8.6 months), respectively. In addition, Kwakman et al. [8] showed an OS of 8.5 months (95% CI, 5.2 months–11.8 months) in patients who had not received all standard therapies prior to TAS-102, compared with 4.7 months (95% CI, 3.6–5.8) for patients who had received all standard treatments.

The majority of patients (80%) included in our study received two or fewer previous lines of therapy. Similarly, in the Wallander et al. [18] study, the subgroup of patients who had received two or fewer prior lines of treatment had a median OS of 7.8 months (CI 95%, 5.3–10.2), conversely patients who had received three or more lines had a median OS of 5.3 months (CI 95%, 3.1–7.5).

Performance status is a well-known negative prognostic factor in advanced disease, particularly in cases of refractoriness to antineoplastic therapy [19,20,21]: ECOG PS was significantly associated with increased mortality at the univariate analysis in our cohort, with a 73% increased risk of death for values of 1 and more than doubled for values of 2 or more, if compared to values of 0. Our results highlight the importance of a careful evaluation of the risk-benefit ratio associated with the administration of antineoplastic drugs in patients with advanced stages of the disease.

Our results show a slight gain in median PFS (3 months, 95% CI 3–4) compared with the RECOURSE trial (2 months, 95% CI 1.9–2.1) [4] but in line with findings from the Japanese registrational TERRA trial [22], the PRECONNECT study [6] and the Sforza et al. study [14], all reporting a median PFS of 2.8 months, and the Stavraka et al. study (3.3 months) [9]. Subgroup analysis of the PRECONNECT study [6] showed a median PFS of 3.1 months (95% CI 2.8–3.5) in patients who had received two or fewer lines prior to TAS-102 [23]. The higher number of patients treated with trifluridine/tipiracil as third-line or earlier therapy in our study (80% of our population vs. 18% of the RECOURSE population) could be a possible explanation for the slight benefit in PFS [4]. Another possible explanation was the different calculation of PFS: we used clinical disease progression and not only radiological progression alone, which usually precedes clinical progression. Indeed, CT scans were generally performed every 3–4 cycles in our cohort and every two cycles in the RECOURSE trial [4].

To evaluate the effectiveness of TAS-102 in elderly patients we compared OS and PFS in patients aged 70, 75, and 80 years and above, with OS and PFS in younger patients. Our data show the absence of an age-related worsening of trifluridine/tipiracil activity/benefit. Indeed, we showed a significant gain of PFS in patients aged 80 or more compared to younger patients (5 months vs. 3 months; *p* = 0.042), suggesting a modest clinically relevant prognostic role of age compared to disease biology in patients with mCRC

The toxicity profile in our cohort was acceptable, with the prevalence of hematologic AEs, mainly in the form of neutropenia among grade G3-G4 toxicities and with a low percentage of patients forced to discontinue treatment due to AEs (2% in our population, 4% in the RECOURSE trial) [4]. A lower frequency of adverse events (88% vs. 98% in the RECOURSE trial), and a lower frequency of major hematologic and non-hematological toxicities were found, except for asthenia, while the frequency of febrile neutropenia was comparable in the two studies [4]. The hematologic toxicity profile in the PRECONNECT study [6] was similar to our cohort, with a prevalence of neutropenia in 53% of patients (vs. 49% in our population), anemia in 29.6% (vs. 22%) and thrombocytopenia in 9.5% (vs. 6%), such as for asthenia (37.3% in PRECONNECT study, 42% in our work). Kwakman et al. [8] reported gastrointestinal toxicity, nausea in 19% of the case series (vs. 12% in our population), vomiting in 5% (vs. 6% in our population) and diarrhea in 12% (vs. 14% in our population). Moreover, a relevant percentage of our patients (25.6%) were treated with one or more lines of treatment after discontinuation of TAS-102. Overall, our results confirm the tolerability of trifluridine/tipiracil in clinical practice, a setting in which adverse events were generally manageable with drug deferrals, dosage reductions and supportive therapies (granulocyte stimulating factor, transfusions, antiemetic and antidiarrheal drugs).

Our study suggests an acceptable toxicity profile of TAS-102 in patients aged 70, 75 and 80 years and older, being the main differences with younger patients related to a lower frequency of asthenia and a higher risk of neutropenia and febrile neutropenia, which lead to discontinuation of trifluridine/tipiracil treatment in only one patient. A Japanese post-marketing surveillance study found advanced age as a risk factor for the development of G3-G4 toxicities (HR for any toxicity: 2.3, 95% CI 1.6–3.4; HR for hematologic toxicity: 2.3, 95% CI 1.6–3.4) [24].

The toxicity profile in patients with heart disease was similar to the overall study population. No cardiopathic patient reported cardiovascular toxicity, compared to only one patient reporting hypotension in the overall cohort. Petrelli et al. [5] found a frequency of cardiotoxicity of 0.5% (between 2 and 40 times lower than the toxicity of 5-fluorouracil) and identified the mechanism of action of trifluridine within the DNA as the main justification for the different cardiotoxicity profile of TAS-102 compared to classical fluoropyrimidine analogs. TAS-102 may be a viable alternative in patients with a history of heart disease and in patients who have undergone cardiotoxicity during treatment with 5-fluorouracil or capecitabine, in which this type of toxicity is more frequent [25,26,27].

Among the three HIV+ patients treated with TAS-102, one presented a severe AE (i.e., febrile neutropenia), one had no AE, and one reported neutropenia and diarrhea (grading not available). Although none of these patients discontinued treatment due to toxicities, the limited size of the sample, the heterogeneity of the results and the absence of further solid data in the literature do not allow us to draw any conclusion on the safety of trifluridine/tipiracil in HIV+ patients.

Only one patient within our population had a *DPYD* gene polymorphism (IVS10 rs75017182): the patient, who received treatment with the reduced dosage because of pre-existing impaired renal function, did not experience severe hematologic or gastrointestinal AEs and did not discontinue treatment because of toxicities. Similar results emerged in six patients described by Schouten et al. In their report, none of the patients experienced severe hematologic and gastrointestinal toxicities, nor required a dose reduction of TAS-102 [13]. Despite these results suggest the possibility of a safe administration of trifluridine/tipiracil in case of *DPYD* polymorphisms, further studies are warranted.

Our study has several limits. First, the retrospective design has per definition several intrinsic potential biases. In particular, several factors may influence (i.e., selection bias) physicians’ decisions regarding the choice of antineoplastic treatment and, therefore, may bias our results. Second, there was a non-homogenous reporting of grade toxicities among recruiting centers, which may have affected the assessment of the severity of the adverse effects reported and some clinical data were not available. Third, several subgroups were underrepresented or even not properly identified (e.g., HIV and *DPYD* test was not performed in many patients): as stated before, any definitive conclusion cannot be drawn. Fourth, regarding progression-free survival, the main limitations included the frequency of instrumental re-evaluations, performed in clinical practice every 3–4 cycles instead of two cycles as in clinical trials, and the need to consider clinical as well as radiologic progression, for reasons related to the study design and to the rapid clinical course of the disease; this may have positively influenced the calculation of PFS in this study. The sample size was not calculated. Indeed, the study is unfortunately underpowered for several analyses. However, we analyzed all patients treated with TAS-102 in our seven oncologic centers. Finally, the absence of a multivariate analysis for overall survival did not allow further investigation of the findings of the univariate survival analysis.

In conclusion, our retrospective multicenter cohort study confirms the effectiveness and safety of TAS-102 in patients with pretreated mCRC, suggesting a similar risk-benefit profile in the elderly and in some special subgroups. Future studies should be focused on special subgroups of patients, to confirm that TAS-102 may be offered to a wide range of patients with pretreated mCRC.

## 5. Conclusions

Our retrospective study assessed the efficacy and the safety of TAS-102 in a real-life large population and it highlights the acceptability of the toxicity profile of TAS-102 treatment also in elderly patients and in special subgroups. Our results are in line with the available literature and reinforce the previous findings.

## Figures and Tables

**Figure 1 cancers-15-03465-f001:**
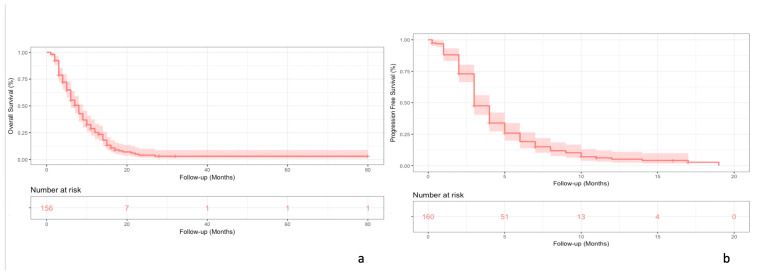
Kaplan–Meier curves for OS (**a**) and PFS (**b**) in the overall population.

**Figure 2 cancers-15-03465-f002:**
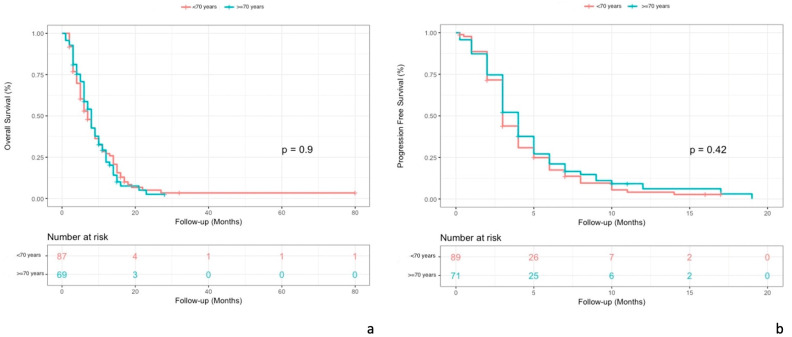
Comparison of Kaplan–Meier curves for OS (**a**) and PFS (**b**) in patient populations aged 70 years and older and patients younger than 70 years.

**Figure 3 cancers-15-03465-f003:**
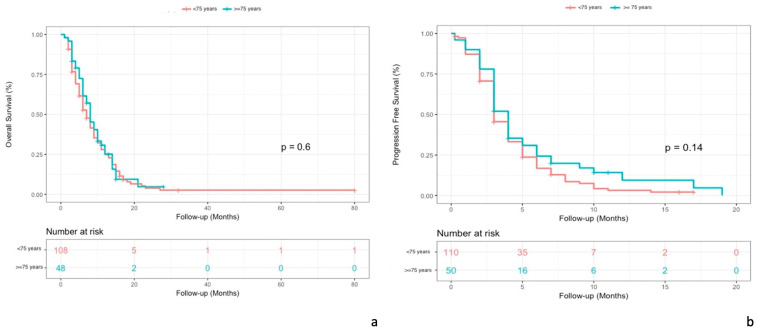
Comparison of Kaplan–Meier curves for OS (**a**) and PFS (**b**) in patient populations aged 75 years and older and patients younger than 75 years.

**Figure 4 cancers-15-03465-f004:**
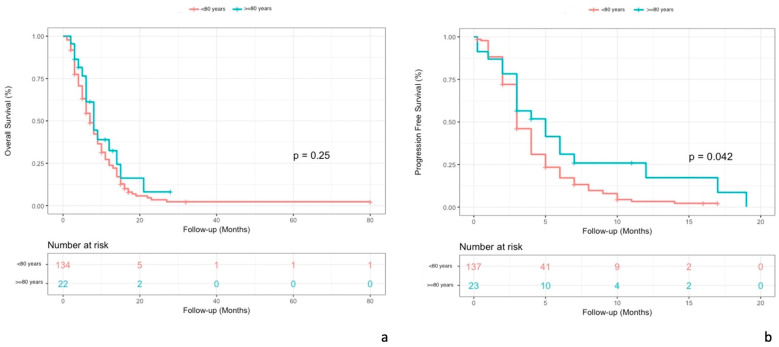
Comparison of Kaplan–Meier curves for OS (**a**) and PFS (**b**) in patient populations aged 80 years and older and patients younger than 80 years.

**Table 1 cancers-15-03465-t001:** Baseline characteristics of the population.

	Value	Patients (n.)	Patients (%)
**Age**			
**Median**	68 years		
**Range**	38–89 years		
**<70 years**		89	56%
**≥70 years**		71	44%
**<75 years**		110	69%
**≥75 years**		50	31%
**<80 years**		137	86%
**≥80 years**		23	14%
**Gender**			
**Male**		103	64%
**Female**		57	36%
**ECOG PS**			
**0**		47	29%
**1**		100	63%
**2+**		13	8%
**Sidedness**			
**Right colon**		45	28%
**Left colon**		71	44%
**Rectum**		44	28%
**RAS**			
**Mutated**		90	56%
**Wild Type**		63	40%
**ND**		7	4%
**BRAF**			
**Mutated**		5	3%
**Wild Type**		134	84%
**ND**		21	13%
**MSS/MSI**			
**MSS**		102	64%
**MSI**		4	3%
**ND**		54	33%
**Time between diagnosis and metastases**			
**>18 months**		113	71%
**<=18 months**		38	24%
**ND**		9	6%
**Metastatic sites n.**			
**1**		44	27%
**2 or more**		116	73%
**Previous treatment lines**			
**0**		2	1%
**1**		11	7%
**2**		115	72%
**3**		25	16%
**4 or more**		7	4%
**Previous therapies**			
**Fluoropirimidines**		158	99%
**Oxaliplatin**		146	91%
**Irinotecan**		147	92%
**Bevacizumab**		122	76%
**Aflibercept**		30	19%
**Anti-EGFR**		53	33%
**Regorafenib**		10	6%
**Cardiac disease**			
**Yes**		19	12%
**No**		134	84%
**ND**		7	4%
** *DPYD* **			
**Mutated**		1	<1%
**Wild Type**		21	13%
**ND**		138	86%
**HIV**			
**HIV+**		3	2%
**HIV-**		108	68%
**ND**		49	31%
**Staging at diagnosis**			
**Stage IV**		85	53%
**Stages I–II–III**		75	47%
**TOTAL**		160	

**Table 2 cancers-15-03465-t002:** Survival analysis-univariate Cox model.

		HR	95% CI	*p*-Value
**Gender**	Female	1.00	reference	
	Male	1.02	0.71–1.45	0.92
**Age**		0.99	0.98–.02	0.93
	<70	1.00	reference	
	>=70	1.02	0.73–1.44	0.91
	<75	1.00	reference	
	>=75	0.91	0.63–1.32	0.62
	<80	1.00	reference	
	>=80	0.75	0.45–1.27	0.28
**Sidedness**	Rectum	1.00	reference	
	Right colon	0.98	0.62–1.53	0.91
	Left colon	0.82	0.54–1.24	0.35
**PS ECOG**	0	1.00	reference	
	1	1.73	1.17–2.55	**0.01**
	>=2	2.12	1.10–4.09	**0.02**
**RAS** **status**	Wild type	1.00	reference	
	Mutated	0.97	0.69–1.37	0.86
**BRAF** **status**	Wild type	1.00	reference	
	Mutated	1.39	0.51–3.78	0.52
**MSS/MSI**	MSS	1.00	reference	
	MSI	0.66	0.21–2.12	0.49
**Staging at diagnosis**	I-II-III	1.00	reference	
	IV	1.26	0.89–1.78	0.19
**Previous treatment lines**	<=2	1.00	reference	
	>2	0.92	0.59–1.43	0.71
**Regorafenib**	No	1.00	reference	
	Sì	0.84	0.44–1.61	0.60
**Metastatic sites n.**	1	1.00	reference	
	>1	1.38	0.93–2.04	0.11
**Time between metastases and TAS-102 treatment**	<=18 months	1.00	reference	
	>18 months	0.93	0.63–1.38	0.73
**Liver metastases**	No	1.00	reference	
	Yes	1.37	0.92–2.05	0.13
**Lung metastases**	No	1.00	reference	
	Yes	0.86	0.61–1.20	0.37
**Peritoneal metastases**	No	1.00	reference	
	Yes	1.25	0.84–1.87	0.27

**Table 3 cancers-15-03465-t003:** Adverse events recorded in the population.

Adverse EvenT (AE)	Any Grade	G1–G2	G3–G4	N. Grade NA
**Leukopenia**	46 (29%)	15 (9%)	4 (3%)	27 (17%)
**Neutropenia**	78 (49%)	14 (9%)	30 (19%)	34 (21%)
**Anemia**	36 (23%)	16 (10%)	7 (4%)	13 (8%)
**Thrombocytopenia**	5 (3%)	4 (3%)	0	1 (<1%)
**Febrile neutropenia**	5 (3%)	0	5 (3%)	0
**Asthenia**	67 (42%)	21 (13%)	3 (2%)	43 (27%)
**Nausea**	19 (12%)	4 (3%)	0	15 (9%)
**Vomiting**	10 (6%)	2 (1%)	0	8 (5%)
**Diarrhea**	23 (14%)	11 (7%)	1 (<1%)	11 (7%)
**Hepatotoxicity**	6 (4%)	0	1 (<1%)	5 (3%)
**Nephrotoxicity (>crea)**	1 (<1%)	0	0	1 (<1%)
**Skin rash**	2 (1%)	0	0	2 (1%)
**Itching**	1 (<1%)	0	0	1 (<1%)
**Mucositis**	3 (2%)	0	0	3 (2%)
**Declivuos edema**	2 (1%)	0	0	2 (1%)
**Gastrointestinal bleeding**	2 (1%)	0	0	2 (1%)
**Alopecia**	1 (<1%)	1 (<1%)	0	0
**Anorexia**	1 (<1%)	0	0	1 (<1%)
**Dysgeusia**	1 (<1%)	0	0	1 (<1%)
**Pneumonia**	1 (<1%)	0	0	1 (<1%)
**Hypoglycemia**	1 (<1%)	0	0	1 (<1%)
**Constipation**	2 (1%)	0	0	2 (1%)
**Paresthesias**	1 (<1%)	0	0	1 (<1%)
**Headhache**	1 (<1%)	0	0	1 (<1%)
**Hypotension**	1 (<1%)	0	0	1 (<1%)
**No AE**	19			

**Table 4 cancers-15-03465-t004:** Adverse events in elderly patients and comparison with younger patients, expressed in number and (% of the subpopulation). * 34 patients with unknown grading excluded from the analysis regarding G3–G4 Neutropenia.

	Neutropenia		G3–G4 Neutropenia *		Leukopenia		Anemia		Thrombocytopenia		Febrile Neutropenia		Asthenia		Diarrhea		Nausea		Vomiting	
	No	Yes	No/G1–G2	G3–G4	No	Yes	No	Yes	No	Yes	No	Yes	No	Yes	No	Yes	No	Yes	No	Yes
<70 years	48 (53.9)	41 (46.1)	55 (77.5)	16 (22.5)	60 (67.4)	29 (32.3)	67 (75.3)	22 (24.7)	84 (94.4)	5 (5.6)	88 (98.9)	1 (1.1)	49 (55.1)	40 (44.9)	75 (84.3)	14 (15.7)	75 (84.3)	14 (15.7)	81 (91.0)	8 (9.0)
>=70 years	34 (47.9)	37 (52.1)	41 (74.6)	14 (25.4)	54 (76.1)	17 (23.9)	57 (80.3)	14 (19.7)	71 (100.0)	0 (0.0)	67 (94.4)	4 (5.6)	44 (62.0)	27 (38.0)	62 (87.3)	9 (12.7)	66 (93.0)	5 (7.0)	69 (97.2)	2 (2.8)
** *p* ** **-value**	0.45		0.70		0.23		0.45		0.07		0.17		0.38		0.58		0.09		0.19	
**<75 years**	63 (57.3)	47 (42.7)	72 (80.0)	18 (20.0)	78 (70.9)	32 (29.1)	84 (76.4)	26 (23.6)	105 (95.5)	5 (4.6)	109 (99.1)	1 (0.9)	58 (52.7)	52 (47.3)	95 (86.4)	15 (13.6)	95 (86.4)	15 (13.6)	102 (92.7)	8 (7.3)
>=75 years	19 (38.0)	31 (62.0)	24 (66.7)	12 (33.3)	36 (72.0)	14 (28.0)	40 (80.0)	10 (20.0)	50 (100.0)	0 (0.0)	46 (92.0)	4 (8.0)	35 (70.0)	15 (30.0)	42 (84.0)	8 (16.0)	46 (92.0)	4 (8.0)	48 (96.0)	2 (4.0)
** *p* ** **-value**	**0.02**		0.11		0.89		0.61		0.33		**0.04**		**0.04**		0.69		0.31		0.73	
**<80 years**	73 (53.3)	64 (46.7)	86 (79.6)	22 (20.4)	99 (72.3)	38 (27.7)	106 (77.4)	31 (22.6)	132 (96.4)	5 (3.7)	134 (97.8)	3 (2.2)	75 (54.7)	62 (45.3)	118 (86.2)	19 (13.9)	121 (88.3)	16 (11.7)	129 (94.2)	8 (5.8)
>=80 years	9 (39.1)	14 (60.9)	10 (55.6)	8 (44.4)	15 (65.2)	8 (34.8)	18 (78.3)	5 (21.7)	23 (100.0)	0 (0.0)	21 (91.3)	2 (8.7)	18 (78.3)	5 (21.7)	19 (82.6)	4 (17.4)	20 (87.0)	3 (13.0)	21 (91.3)	2 (8.7)
** *p* ** **-value**	0.21		**0.03**		0.49		0.92		0.99		0.15		0.03		0.75		0.74		0.64	

## Data Availability

The data presented in this study are available in the article.

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
