# Peer review of "Tas-102 for Refractory Metastatic Colorectal Cancer: A Multicenter Retrospective Cohort Study"

_cancers, 2023, doi:10.3390/cancers15133465_

Round 1

Reviewer 1 Report

Conti et al analyzed a real-life Italian cohort treated with TAS-102 for metastatic colorectal cancer. This cohort is interesting because nearly half of the patients had 70 years old or older and 31% were 75 years old or older. The litterature on the subject is quite weak, however it is a daily problem because patients with colorectal cancer are often elderly.  

The progression-free survival and overall survival results are identical to those in the literature, showing maintained efficacy in this population. On the other hand, an altered performans status (1 or 2) has a worse prognosis, which had already been demonstrated for younger patients.

The tolerance of chemotherapy is quite maintained in the elderly, with only in the over 80s a greater risk of neutropenia. The authors also produced data on patients with heart disease, with HIV infection or with DPD deficiency, but on small numbers making it difficult to generalize these results.

The article is well written and well presented, the discussion is developed.

I have no major concerns about this article. I would only recommend an editing of the survival curves by suppressing the term " strata". I was surprised in the tables of the lack of unavailable data, given the retrospective nature of the study, can you confirm this?

The article is well written, minor editing of English language is required

Author Response

Dear Editor,

In behalf of all of the authors I really thank you for your time and your comments.

  • We edited the survival curve as requested, thank you for the suggestion.
  • We confirm that the tables are correct. Unfortunately some data are lacking this represents a limit of the study (we specified it in the main text, line 427).

Best regards,

Reviewer 2 Report

Dear Authors:

This retrospective multicenter cohort study aimed to assess the effectiveness and safety of TAS-102 in a real-life context. In particular, they planned to assess the effectiveness and safety of TAS-102 in elderly patients and in some special subgroups, such as patients with concomitant heart diseases, DPYD polymorphisms or HIV positive. Although an increasing number of elderly patients required cancer treatments in advanced lines, these patients were often underrepresented in clinical trials. However, this research gave a comprehensive talk about the effectiveness and safety of TAS-102 in these special population. 

It might be better if the authors could trouble to consider the following issues according to the actual situations:

1.     It might be better for the authors to increase the sample size of this research.

2.     A brief introduction to patients with concomitant heart diseases, DPYD polymorphisms or HIV positive might be better to be included in the research background.

Author Response

Dear Editor,

In behalf of all of the authors I really thank you for your revision.

  • At the moment it is impossible for us to increase the sample size of our research because it would imply to find other patients from other oncological centers (we included all of the patients treated with TAS 102 in seven oncological units), perform a new statistical analysis and rewrite the paper completely. I hope that you and the other editors understand it.

  • We completed the introduction as requested (line 62, 64-68,71, 86,92).

Best regards,

Round 2

Reviewer 2 Report

Dear Authors:

This retrospective multicenter cohort study aimed to assess the effectiveness and safety of TAS-102 in a real-life context. In particular, they planned to assess the effectiveness and safety of TAS-102 in elderly patients and in some special subgroups, such as patients with concomitant heart diseases, DPYD polymorphisms or HIV positive. Although an increasing number of elderly patients required cancer treatments in advanced lines, these patients were often underrepresented in clinical trials. However, this research gave a comprehensive talk about the effectiveness and safety of TAS-102 in these special population. However, It might be better to increase the sample size of this research.